# Serum Biomarkers in Connective Tissue Disease-Associated Pulmonary Arterial Hypertension

**DOI:** 10.3390/ijms24044178

**Published:** 2023-02-20

**Authors:** Beatrice Moccaldi, Laura De Michieli, Marco Binda, Giulia Famoso, Roberto Depascale, Martina Perazzolo Marra, Andrea Doria, Elisabetta Zanatta

**Affiliations:** 1Rheumatology Unit, Department of Medicine-DIMED, Padova University Hospital, 35128 Padova, Italy; 2Department of Cardiac, Thoracic, Vascular Sciences and Public Health, Padova University Hospital, 35128 Padova, Italy

**Keywords:** pulmonary arterial hypertension, biomarkers, connective tissue disease, pathogenesis

## Abstract

Pulmonary arterial hypertension (PAH) is a life-threatening complication of connective tissue diseases (CTDs) characterised by increased pulmonary arterial pressure and pulmonary vascular resistance. CTD-PAH is the result of a complex interplay among endothelial dysfunction and vascular remodelling, autoimmunity and inflammatory changes, ultimately leading to right heart dysfunction and failure. Due to the non-specific nature of the early symptoms and the lack of consensus on screening strategies—except for systemic sclerosis, with a yearly transthoracic echocardiography as recommended—CTD-PAH is often diagnosed at an advanced stage, when the pulmonary vessels are irreversibly damaged. According to the current guidelines, right heart catheterisation is the gold standard for the diagnosis of PAH; however, this technique is invasive, and may not be available in non-referral centres. Hence, there is a need for non-invasive tools to improve the early diagnosis and disease monitoring of CTD-PAH. Novel serum biomarkers may be an effective solution to this issue, as their detection is non-invasive, has a low cost and is reproducible. Our review aims to describe some of the most promising circulating biomarkers of CTD-PAH, classified according to their role in the pathophysiology of the disease.

## 1. Introduction

Pulmonary hypertension (PH) is a haemodynamic state characterised by elevated blood pressures in the pulmonary vessels. According to the latest guidelines of the European Society of Cardiology (ESC) and the European Respiratory Society (ESR) [1], PH is defined by a mean pulmonary arterial pressure (mPAP) that is greater than 20 mmHg assessed by right heart catheterisation (RHC); compared to the 2015 guidelines [2], this threshold has been lowered (i.e., from 25 to 20 mmHg) based on studies to assess the normal range of pulmonary pressures in healthy subjects [3,4,5] and the poor outcome in patients with an mPAP that is greater than 20 mmHg [6,7,8]. PH is currently classified into five groups based on aetiology and pathogenesis. Group 1 PH refers to pulmonary arterial hypertension (PAH), which is a pre-capillary condition characterised by the vascular remodelling of the pulmonary arterioles, leading to increased pulmonary vascular resistance (PVR), and ultimately, to right heart failure; RHC shows increased PVR (i.e., >2 Woods Units) and normal pressures in the left atrium (i.e., pulmonary artery wedge pressure, PAWP, ≤15 mmHg). PAH may be idiopathic (IPAH), inherited, induced by drugs or toxins or secondary to an underlying disease. According to the latest classification, group I PAH also includes PAH with features of venous/capillary involvement (pulmonary veno-occlusive disease (PVOD)/pulmonary capillary haemangiomatosis (PCH)) and persistent PH of the newborn. PAH is a rare condition with a prevalence of approximately 48–55/1,000,000 people in Western countries [9]. PAH associated with connective tissue disease (CTD, 10–30%) [10] is the most common subtype after IPAH. Systemic sclerosis (SSc)-associated PAH accounts for 75% of all CTD-PAH cases; it affects 8–15% of SSc patients over the course of the disease [11,12] and carries a worse prognosis than other forms of PAH do, including IPAH [13]. Moreover, PAH may be a complication, albeit less frequently, of any other CTD, such as systemic lupus erythematosus (SLE, 1–5% of all cases [14]), mixed connective tissue disease (MCTD), and more rarely, primary Sjögren’s syndrome (pSS), idiopathic inflammatory myopathies (IIMs) and rheumatoid arthritis (RA) [15,16,17]. However, it bears noting that any PH group may be detected in patients with CTDs, each carrying specific prognostic and therapeutic implications, thus highlighting the importance of a thorough differential diagnosis once PH is suspected in these patients [15,16,17,18,19,20]. Diagnosing PAH in patients with CTDs may be challenging due to the non-specific nature of the early symptoms and the lack of screening strategies (except for SSc, with a yearly transthoracic echocardiography recommended) [21,22]; therefore, PAH is often diagnosed in an advanced stage when the pulmonary vessels are irreversibly damaged. In addition, the gold standard to diagnose PAH is RHC, an invasive procedure that is often unavailable in non-referral centres. Hence, there is the need to explore non-invasive tools for disease monitoring and early detection of PAH in CTD patients. The DETECT algorithm was developed to provide a multi-modality screening tool for PAH in patients with SSc: it combines elements from clinical examination, electrocardiography, pulmonary function tests, blood tests and echocardiography into a composite score with high sensitivity (95%), but relatively low specificity (48%), for the presence of SSc-PAH [23]. On the other hand, serum biomarkers of PAH may constitute a non-invasive, feasible and reproducible diagnostic tool in both SSc and other CTDs. A biomarker is defined as “a characteristic that is measured objectively and evaluated as an indicator of normal biological processes, pathogenic processes or pharmacological responses to a therapeutic intervention” [24]; in other words, a serum biomarker is a molecular product whose presence may be detected in the blood due to a specific pathophysiological process or therapeutic intervention. Several potential diagnostic and prognostic biomarkers of CTD-PAH have been identified and may be used in the near future in clinical practice. Our review aims to describe some of the more promising candidates, categorized according to their role in the complex pathophysiology of CTD-PAH, which includes endothelial dysfunction and vascular remodelling, autoimmunity, inflammation and cardiac dysfunction. The main biomarkers and the associated pathogenetic pathways are summarized in Figure 1.

In the early stages of the disease, the combined action of inflammatory cytokines, vasoactive mediators, angiogenic factors, products of autoimmunity and endothelial-to-mesenchymal transition contributes to vascular remodelling, resulting in increased pulmonary vascular resistances. Subsequently, hypoxia develops in the lungs and in the peripheral tissues, and the hemodynamic changes incurred as a result lead to right ventricle dysfunction and failure.

## 2. Endothelial Dysfunction

Endothelial dysfunction plays a pivotal role in the development of CTD-PAH. It stems from the downregulation of vasodilators and upregulation of vasoconstrictors and proliferative mediators, which affect the vascular tone and promote vascular remodelling; platelet activation and impaired coagulative status further contribute to this process. The main biomarkers of endothelial dysfunction in CTD-PAH are listed in Table 1.

### 2.1. Inbalance between Vasoactive Mediators

In CTD-PAH, activated pulmonary endothelial cells increase the production of vasoconstrictors and reduce the production of and responsiveness to vasodilators.

#### 2.1.1. Endothelin-1 (ET-1)

ET-1, produced by vascular endothelial cells, is a potent vasoconstrictor of the pulmonary circulation via the activation of two endothelin receptor isoforms, type A (ETAR) and type B (ETBR), expressed on pulmonary arterial smooth muscle cells (SMCs) [25,26]. ETBR activation on endothelial cells induces ET-1 clearance and vasodilation through the release of nitric oxide and prostaglandins, ultimately leading to vasodilation, and thus, balancing the vasoconstrictive effect of ET-1 [25]. In addition to its vasoactive properties, ET-1 plays an important role in modulating several inflammation pathways, both directly, by inducing the migration of neutrophils and monocytes [26,27], and indirectly, by increasing the production of inflammatory cytokines [28] and adhesion molecules in different cell types [29,30]. A recent study on ETBR immunodynamics highlighted the presence of a regulatory feedback of the effect of ET-1 on immune cells: ETBR deficiency in engineered mice led to PAH, pulmonary vascular hyperresponsiveness and right ventricular hypertrophy with lymphocytic alveolitis and marked lymphocytic perivascular infiltrate, suggesting that ETBR-mediated signalling pathways may have an immunomodulatory effect [31]. Increased ET-1 levels have been found in the serum and in the lungs of IPAH patients [32,33]. Several studies reported elevated plasma ET-1 levels in SSc-PAH patients compared to those of healthy controls [34,35]; other authors found a correlation between the ET-1 levels and capillary size on nailfold videocapillaroscopy (NVC) [36], echocardiographic signs of impaired right ventricular function [32] and responses to the endothelin receptor antagonist bosentan in patients with SSc-PAH [37]. Elevated serum ET-1 levels have been found also in patients with SLE compared to those of healthy controls [38], and they may correlate with the development of PAH [39]. Notably, circulating ET-1 levels vary based on its clearance by ETBR on endothelial cells, and thus, may not reflect the actual ET-1 expression levels in the lungs.

#### 2.1.2. Nitric Oxide (NO) Pathway

NO is a potent pulmonary vasodilator, as well as an inhibitor of platelet activation and SMCs proliferation [40]. NO is produced in the endothelial cells by activated NO synthase (NOS): there are three known isoforms of this enzyme, namely, neuronal NOS (nNOS), endothelial NOS (eNOS) and inducible NOS (iNOS), which is activated during inflammation. In SSc-PAH, circulating NO levels have been shown to decrease after 24 weeks of therapy with pulmonary vasodilators (e.g., bosentan), which is possibly related to the treatment response [37].

Asymmetric dimethylarginine (ADMA) is an endogenous inhibitor of eNOS. In a small cohort of SSc-PAH patients (*n* = 15), ADMA levels were significantly associated with PAH, and the combination of ADMA levels ≥ 0.7 ng/mL and N-terminal pro B-type natriuretic peptide (NT-proBNP) ≥ 210 ng/mL showed 100% sensitivity and 90% specificity for the presence of SSc-PAH, respectively [41]; another study found a negative correlation between serum ADMA levels and the 6-minute walking distance (6MWD) in patients with SSc-PAH [42]. However, some other studies did not find any significant correlation between ADMA levels and echocardiographic signs of PH [36,43].

### 2.2. Vascular Remodelling

#### 2.2.1. Endothelial-to-Mesenchymal Transition

The progressive remodelling of small- to medium-sized pulmonary arteries is crucial in PAH pathogenesis and largely depends on myofibroblasts activation. Myofibroblasts are a unique population of mesenchymal cells displaying a marked profibrotic cellular phenotype characterised by an increased production of type I and type III collagens and the expression of α-smooth muscle actin (α-SMA); activated myofibroblasts have migratory and contractile properties, perform an invasive behaviour and have increased resistance to apoptosis, which allows them to migrate and induce increased tissue stiffness. Although myofibroblasts may derive from many different cell lines, in SSc-PAH, a central role is attributed to endothelial-to-mesenchymal transition, a molecular process wherein endothelial cells progressively lose their tissue-specific markers and acquire a myofibroblastic phenotype [44]. The evidence that this process is involved in the pathogenesis of vascular remodelling in SSc-PAH stems from several studies, which have demonstrated the presence of transitional endothelial–mesenchymal cells in the pulmonary vessels of affected patients [44,45]. Endothelial-to-mesenchymal transition appears to be triggered mainly via the transforming growth factor (TGF)-β-superfamily-signalling pathway [46,47,48], which activates the intracellular transcription factors responsible for the phenotype switch. Alongside TGF-β, other inflammatory cytokines, such as tumour necrosis factor (TNF)-α and interleukin (IL)-1β [49], may also contribute to the induction of the transdifferentiation of the endothelial cells. Conversely, the bone morphogenic protein-receptor 2 (BMPR2)-driven molecular signalling pathway balances the action of TGF-β through the inhibition of SMCs proliferation and the activation of eNOS, resulting in increased vasodilation [50]. Several molecular products of endothelial-to-mesenchymal transition and myofibroblast activation have been studied as potential biomarkers of SSc-PAH.

Growth Differentiation Factor-15 (GDF-15) is a member of the TGF-β superfamily; increased levels have been reported in cases of heart failure, atherosclerosis, endothelial dysfunction and diabetes [51], and it is a prognostic biomarker in IPAH [52]. Meadows et al. [53] found that the plasma and tissue levels of GDF-15 were significantly higher in SSc-PAH vs. those in SSc without PAH, IPAH and in healthy controls. Within the SSc subgroup, the GDF-15 levels correlated with echocardiographic right ventricular systolic pressure and the NT-proBNP levels. In the ROC curve, GDF-15 had good discriminatory power for the identification of SSc-PAH, with an optimal cut-off value of 125 pg/mL (93% sensitivity and 88% specificity). In addition, patients with GDF-15 levels below this threshold had significantly better survival rates. This finding was corroborated in a recent study, which found higher GDF-15 levels in SSc patients vs. those in healthy controls and a significant association between elevated GDF-15 and PH in SSc patients [54].

Endoglin (Eng) is a transmembrane protein expressed in endothelial cells, where it promotes TGF-β signalling amplification. High serum Eng levels have been found in IPAH patients [55]; Coral-Alvarado et al. [56] investigated circulating Eng concentrations in 60 patients with PH, as assessed by echocardiography (20 SSc-PAH, 20 SSc no-PAH and 20 healthy controls). The authors reported higher Eng concentrations in the SSc-PAH group, though only the difference between SSc-PAH vs. healthy controls was statically significant, likely due to the small sample size. No correlation was found between Eng concentration and echocardiographic estimated PH.

Follistatin-like 3 (FSTL3) is a TGF-β-regulated protein that was found in high concentrations in the serum of patients with diffuse cutaneous SSc (dcSSc) [57]. Midkine (MDK) is a novel cytokine that appears to induce fibroproliferation in the lungs through the activation of its receptor Notch2 [58]. Rice et al. [59] found that both FSTL3 and MDK were upregulated in the serum of treatment-naïve patients with PAH and limited cutaneous SSc (lcSSc) vs. those in lcSSc patients without PAH. However, serum FSTL3 and MDK levels did not correlate with each other, nor with the BNP levels. Sera from two validation cohorts were then used to test the discrimination of FSTL3 and MDK for lcSSc-PAH and although some of the patients had already received treatment, a significant increase in both FSTL3 and MDK concentrations was found in lcSSc with PAH vs. those without PAH. The combination of MDK and FSTL3 had an area under the curve (AUC) of 0.85 with 76% sensitivity and 76% specificity in the first validation cohort and an AUC of 0.92 with 91% sensitivity and 80% specificity in the second validation cohort. These findings highlight FSTL3 and MDK as two promising diagnostic biomarkers in patients with lcSSc-PAH.

In a recent study, a proteome analysis was performed on the sera of 31 lcSSc patients (15 with PAH and 16 without PAH) in order to identify a valid biomarker of lcSSc-associated PAH; patients with extensive ILD or who had already been treated for PAH were excluded. Serum levels of chemerin, a pleiotropic protein that exerts its multiple effects on SMCs, inducing their contraction and proliferation, and on endothelial cells, regulating neoangiogenesis, NO production and cell adhesion, were found to be higher in lcSSc patients with PAH vs. in those without it; moreover, in lcSSc-PAH patients, chemerin levels appeared to correlate with PVR, performing as a marker of haemodynamic severity in these patients [60].

Another group performed a similar proteomic analysis on serum samples collected from patients with SSc and PAH (*n* = 77) and SSc without PAH (*n* = 80) from the DETECT study cohort, identifying an eight-protein panel that discriminated PAH patients from non-PAH patients [61]. This panel included two proteins involved in pulmonary vascular remodelling: RAGE (receptor for advanced glycation end products, which has been found to regulate the proliferation of SMCs in vitro via the BMP signalling pathway) and MMP-2 (matrix metalloproteinase-2, involved in the breakdown of the extracellular matrix and the basal membrane).

Other matrix metalloproteinases (MMPs) and tissue inhibitors of metalloproteinases (TIMPs) have been studied as potential SSc-PAH biomarkers. Avouac et al. [62] found that the MMP-10 gene was upregulated in the endothelial cells of SSc-PAH patients and serum pro-MMP10 was increased in SSc-PAH patients vs. SSc patients without PAH and in healthy controls. TIMP-4 may contribute to extracellular matrix deposition in SSc, and its serum levels have been shown to correlate with systolic pulmonary arterial pressure (sPAP) after echocardiography in both SSc-PAH and PH patients in general [63,64]. Other evidence has shown that serum MMPs levels are also increased in SSc patients with interstitial lung disease (ILD) [65,66], whereas TIMPs appear to be more specifically associated with vascular changes [64,67]; however, no studies to date have compared the serum levels of these molecules between patients with SSc-ILD and SSc-PAH.

Osteopontin (OPN) is an extracellular matrix protein that is also involved in the regulation of inflammatory response [68], fibrosis and vascular remodelling [69], and it may be a promising diagnostic and prognostic biomarker in CTD-PAH patients. A recent study measured the serum OPN levels in a cohort of 113 CTD patients (77.9% SSc, 8.8% MCTD, 8.8% overlap syndrome and 4.4% undifferentiated CTD), 16 of whom had a diagnosis of CTD-PAH by RHC. The patients with CTD-PAH had significantly higher serum levels of OPN vs. those of the patients without PAH, but no correlation was observed between OPN levels and mPAP as per RHC [70]. Increased OPN levels have also been reported in patients with SSc-ILD [71]. These findings indicate that increased serum OPN may help to identify a general clinical condition characterised by inflammation and tissue fibrosis, rather than a specific pathological entity.

Uric acid (UA) is a by-product of adenine oxidation and guanine purine metabolism; its serum levels increase when oxidative metabolism is impaired, and hyperuricemia is a common occurrence in tissue hypoxia. Elevated UA levels were found in PAH patients [72], and there was a correlation with the New York Heart Association (NYHA) functional class, mortality and 6MWD [73]; moreover, pulmonary vasodilators significantly decrease the UA in peripheral blood [74]. In SSc patients, UA has been identified as a marker of microvascular damage, showing a good correlation with serum creatinine and estimated sPAP [75]. Elevated UA levels were associated with SSc-PAH diagnosis via RHC and mortality [76]. Due to its potential diagnostic and prognostic value, uric acid is commonly used in clinical practice to detect PAH in SSc patients, as it is a serological item of the composite DETECT score [23].

#### 2.2.2. Impaired Vasculogenesis and Angiogenesis

The progressive intimal proliferation and dysregulated ECM deposition causes the obstruction of the pulmonary vascular lumen, reduced blood flow and tissue hypoxia. This results in the upregulation of hypoxia-dependent genes with increased production of neoangiogenic factors, which attract circulating endothelial cells to the injured tissue. In SSc vasculopathy, the concentration of endothelial progenitor cells in the bloodstream progressively decreases over the course of the disease [77], indicating a pathological response to the neoangiogenic stimuli that results in an impaired and dysfunctional vasculogenesis. Several angiogenic factors have been proposed as potential biomarkers of CTD-PAH, particularly SSc-PAH.

Papaioannou et al. [78] reported a correlation between elevated plasma levels of vascular endothelial growth factor (VEGF) in SSc patients and increased sPAP, though another study was not able to corroborate the association between the elevated VEGF levels and CTD-PAH [79].

Adachi et al. [79] evaluated the serum levels of angiogenic and angiostatic factors in a heterogeneous population of 107 PH patients and 33 healthy controls. The serum levels of VEGF-A165b, an inhibitory splice variant of VEGF-A [80], were significantly higher in the CTD-PAH and congenital heart disease-PAH groups than they were in the controls. Serum placental growth factor (sPlGF) levels were higher in patients with CTD-PAH vs. in those with IPAH and healthy controls, and there was a correlation between sPlGF and 6MWD, BNP levels and the World Health Organisation functional class (WHO-FC).

Increased serum levels of endostatin (ES), a potent angiostatic factor, were found in CTD-PAH patients vs. those in IPAH patients; ES was also significantly higher in CTD-PAH than it was in porto-pulmonary hypertension and appeared to correlate with 6MWD, BNP levels, WHO-FC, PVR and mixed venous oxygen saturation. Other studies have confirmed the increased plasma ES levels in SSc patients, with an association with PAH and mortality [81]; there have also been reports of an association between ES levels and exercise capacity, WHO-FC and haemodynamics in PAH patients [82]. Moreover, McMahan et al. [83] found significantly increased plasma levels of PlGF and VEGF-receptor 1 (i.e., FLT-1, a decoy receptor that binds to free VEGF-A, VEGF-B and PlGF, thus downregulating their activity) in SSc patients complicated by PH vs. those in SSc patients without PAH, but no further information was given regarding the concentration of these biomarkers in the different PH subtypes in SSc.

Finally, neuropilin-1, another molecule that interacts with VEGF family members to induce angiogenesis in endothelial cells, is part of the eight-protein panel for the detection of SSc-PAH stemming from the proteomic analysis from the DETECT cohort [61].

### 2.3. Platelet Activation

In SSc, activated endothelial cells produce adhesion molecules that promote platelet aggregation and the activation of the coagulation cascade; activated platelets then promote coagulation, enhance vasoconstriction and directly stimulate the activation of fibroblasts and the transdifferentiation of endothelial cells into myofibroblasts.

The Von Willebrand factor (vWF) plays a crucial role in the coagulation cascade as a carrier of coagulation factor VIII. Elevated levels of vWF were found in IPAH and in patients with chronic thromboembolic pulmonary hypertension (CTEPH) [84], as well as lcSSc patients with an increased risk for PH [85,86]. However, Iannone et al. [87] observed no difference in the vWF levels between patients with SSc vs. SSc-PAH. vWF levels were also found to be higher in MCTD-PAH patients vs. in MCTD patients without PAH [88].

Thrombomodulin (TM) is a glycoprotein involved in the activation of protein C and in the modulation of thrombin activity; its serum levels are increased in conditions associated with endothelial damage [89]. Stratton et al. [90] found increased plasma levels of soluble TM in SSc-PAH patients vs. those of SSc patients without PH and healthy controls; in a cohort of MCTD patients, the soluble TM level was higher in patients with PAH vs. that of patients without PAH [88]. It bears noting that one study found lower levels of circulating TM in patients with IPAH and PAH due to Eisenmenger syndrome vs. those of healthy controls; this difference with SSc-PAH may be due to a different pathogenesis within the group 1 PAH subtypes [91].
ijms-24-04178-t001_Table 1Table 1Biomarkers of endothelial dysfunction in CTD-PAH.Molecular Mechanisms of Endothelial DysfunctionBiomarkers**Imbalance between vasoactive mediators**ET-1 [32,34,35,36,37,39]NO [37]ADMA [41,42]**Endothelial-to-mesenchymal transition**GDF-15 [53,54]Eng [56]FSTL-3, MDK [59]Chemerin [60]RAGE, MMP2 [61]proMMP-10 [62]TIMP-4 [63,64]OPN [70]UA [23,75,76]**Impaired neoangiogenesis**VEGF [78]VEGF-A165B [79]PlGF [79]ES [79,81]FLT-1 [83]NP-1 [60]**Platelet activation**vWF [85,86,88] TM [88,90]ADMA, asymmetric dimethylarginine; Eng, endoglin; ES, endostatin; ET-1, endothelin-1; FLT-1, PlGF and VEGF receptor-1; FSTL-3, follistatin-like 3; GDF-15, growth differentiation factor-15; MDK, midkine; MMP, matrix metalloproteinase; NO, nitric oxide; NP-1, neuropilin-1; OPN, osteopontin; PlGF, placental growth factor; RAGE, receptor for advanced glycation end products; TIMP, tissue inhibitor of metalloproteinase; TM, thrombomodulin; UA, uric acid; VEGF, vascular endothelial growth factor; vWF, von Willebrand factor.


## 3. Autoimmunity

The pathogenesis of CTD-PAH is complex: fibrotic and vascular changes are a key part of the process, but several autoantibodies have been found to be associated with PAH in CTD patients, suggesting a role for autoimmunity in establishing the disease (Table 2).

### 3.1. Disease-Specific Antibodies: Autoantibodies Specific for CTDs, Which May Be Associated with the Risk of PAH, but Do Not Have a Proven Role in the Pathogenesis of CTD-PAH

#### 3.1.1. Systemic Sclerosis

Recent data have shown that over 80% of SSc-PAH patients have positive antinuclear antibodies (ANA) [92]. Among the SSc-specific antibodies, anti-centromere antibodies (ACAs) show the strongest association with the development of PAH; in a systematic review that analysed 80 cohorts of SSc-patients, ACAs had a prevalence of 45% in those affected by PAH [92], while in the DETECT cohort, ACAs were found to be strongly associated with PAH, and positive ACA was included in the algorithm for identifying patients with SSc-PAH [23]. However, positive ACA does not appear to have prognostic value in the context of SSc-PAH [93]. Recently, a multicentric study group identified a subset of ACA-positive patients whose anti-centromere protein A (anti-CENP-A) autoantibodies were specific for the aminoterminal epitope of CENP-A (amino acids 1–7); these specific anti-CENP-A antibodies were called anti-p4.2 antibodies, since they react with peptide 4.2 (p4.2), a peptide isolated in a phage peptide library. Anti-p4.2 antibodies were found to be associated with a diffusing capacity of the lungs for carbon monoxide (DLCO) < 70%, even in the absence of pulmonary fibrosis, suggesting a role for these autoantibodies in predicting the development of PAH vasculopathy in ACA-positive patients [94]. Other SSc-specific autoantibodies (i.e., anti-topoisomerase I and anti-RNA polymerase III) have not shown an association with PAH [92]. Anti-ribonucleoprotein (RNP) antibodies have been found to be associated with PH [95] and PAH in SSc patients [96,97,98]: anti-U1RNP have been detected in 9% patients with SSc-PAH, and approximately 22% of anti-U1RNP-positive SSc patients were affected by PAH. Anti-U3RNP was detected in about 24% of SSc-PAH patients, whereas PAH was found in 25% of SSc patients with positive anti-U3RNP, suggesting that this autoantibody may play a role in the development of PAH [92]. Anti-Th/To autoantibodies also appear to be associated with PAH in SSc; they were found in 25% of patients with SSc-PAH in eight different cohorts, whereas 33% of the anti-Th/To+ patients had PAH [92]. Interestingly, there have been reports of an association between anti-phospholipid antibodies (aPL) and PAH in patients with SSc, chronic thromboembolic pulmonary disease notwithstanding. Morrisroe et al. [99] tested 940 SSc patients for serum aPL, and demonstrated an association between anti-cardiolipin (aCL) IgG and PAH diagnosed via RHC, with higher titres corresponding to a higher likelihood of PAH (OR 4.60, 95% CI, 1.02–20.8). Nevertheless, aCL, IgM and IgG were also associated with ILD, with aCL IgG emerging as a potential biomarker of ILD with concomitant PH (PH-ILD, group 3) (OR 2.10, 95% CI, 1.05–4.20). Boin et al. [100] identified an association between anti-β2-glycoprotein I (anti-β2GPI) antibodies and echocardiographic signs of PAH in a cohort of 75 SSc-patients (OR 4.8, 95% CI, 1.0–11.4). The data from other SSc cohorts showed a relatively high prevalence of aPL antibodies in patients with PAH [92].

#### 3.1.2. Systemic Lupus Erythematosus

The pulmonary vessels of SLE-PAH patients show immune-complexes-mediated pulmonary vasculitis with mild intimal fibrosis during a histopathologic examination [101], which is in contrast to SSc-PAH, where signs of vascular remodelling predominate over those of inflammation. SLE-PAH may be the result of high disease activity, with autoimmunity and immune complexes formation playing a major role.

Although anti-U1RNP antibodies were not clearly associated with SSc-PAH, they appear to have a strong association with PAH in SLE patients. In a case–control study comparing 84 SLE-PAH patients and 160 SLE patients without PAH, anti-U1RNP antibodies were independently associated with severe PAH and more active SLE [97]. In a multicentric longitudinal study carried out in a cohort of 3624 consecutive SLE patients, a 10-year probability-predictive nomogram for the development of PAH, including clinical variables and three autoantibodies (anti-RNP, anti-SSA and anti-SSB) was developed, confirming the close association between the anti-RNP antibodies and PAH in SLE [102]. Several studies demonstrated an association between aPL antibodies and PAH in SLE. Zuily et al. [103] performed a meta-analysis of 31 studies comprising about 4480 patients with SLE and found that SLE patients with positive aPL antibodies had a higher prevalence of all-cause PH (12.3%) vs. those without aPL antibodies (7.3%) (OR 2.28, 95% CI, 1.65–3.15), in particular, the risk of PAH was increased in the former group (OR 2.62, 95% CI, 1.11–6.15). The strongest association was between PH diagnosed by echocardiography and aCL antibodies [104,105], although Houman et al. [106] found a higher frequency of PH diagnosed by echocardiography also in patients with positive anti-β2GPI antibodies. It is worth mentioning that other causes of PH—including CTEPH, which may be associated with aPL antibodies [107]—were excluded in one-third of the studies analysed by Zuily et al. For example, all 51 SLE-PH patients enrolled in the study by Lee et al. [105] underwent a diffusion/perfusion scan to exclude CTEPH. These findings, albeit limited, suggest that vascular remodelling, rather than thrombosis, may be the predominant pathogenetic pathway in aPL-positive SLE patients with PAH, but the role of aPL antibodies in this process is yet to be clarified.

#### 3.1.3. Other CTDs

Not many data are available on immune biomarkers of PAH in other CTDs. Vegh et al. [88] studied a cohort of 179 anti-U1RNP-positive MCTD patients (25 PAH and 154 non-PAH), demonstrating a slightly higher titre, albeit not statistically significant, of anti-U1RNP antibodies in PAH patients, which is in line with what we mentioned earlier in regard to SLE-PAH; however, the histopathological findings from the pulmonary vessels in MCTD-PAH are more similar to those of SSc-PAH patients, with marked intimal fibrosis and mild inflammatory infiltration [101]. A Chinese study evaluated a multicentric cohort of 103 patients with primary SS (pSS) complicated by PAH and compared their clinical and serological features with those of 526 pSS patients without PAH. Anti-SSB (OR 4.095, 95% CI, 2.183–7.681) and anti-U1RNP antibodies (OR 29.518, 95% CI, 6.026–144.600) were identified as independent risk factors for PAH, whereas anti-SSA antibodies and hypergammaglobulinemia were not [108]. In other studies, however, patients with pSS-PAH were found to have higher rates of positive anti-SSA, rheumatoid factor and hypergammaglobulinemia. It bears noting that even though these findings appear to suggest a B-cell mediated process, pulmonary vasculitis is not a histological hallmark in pSS-PAH [101].

There are scant data in the current literature on the serological profile of patients with IIM-PAH; some studies have reported the association between anti-Ku antibodies and PAH in patients with IIMs and in overlap syndromes [109,110].

### 3.2. Functional Antibodies: Autoantibodies That are Not Specific for CTDs, but Whose Pathogenetic Role in the Development of PAH Has Been Demonstrated by In Vitro and/or In Vivo Studies

#### 3.2.1. Vascular Receptor Autoantibodies

Similar to the aforementioned role of ET-1 and its receptors (ETAR and ETBR) in the pathogenesis of PAH, angiotensin II is a potent vasoactive agent that exerts its pleiotropic vasoconstrictor and pro-inflammatory effect by binding to angiotensin receptor type 1 (AT1R). Autoantibodies targeting ETAR (anti-ETAR) and AT1R (anti-AT1R) have been found in patients affected by SSc, with a prevalence of 85%, showing a strong correlation with each other and a cross-reactivity for both receptors [111]. These autoantibodies mimic ET-1 and angiotensin II and activate signal transduction pathways in non-immune and immune cells, as demonstrated by in vitro and in vivo studies [112]; they induce the production of TGF-β, IL-8 and adhesion molecules by endothelial cells, stimulate T-cell chemotaxis and the secretion of IL-8 and C-C motif ligand (CCL) 18 in monocytes [113], and increase type I collagen deposition [114]. Vasculopathic SSc manifestations, such as PAH and digital ulcers [115], were associated with higher titres of both autoantibodies, which appear to predict SSc-related mortality [113]. Moreover, SSc-PAH and other CTD-PAH patients showed a higher frequency of anti-ETAR/anti-AT1R antibodies compared to that of IPAH patients [116]. Guo et al. [117] demonstrated higher titres of anti-ETAR antibodies in SLE patients complicated by PAH vs. those in SLE patients without PAH (41.5% vs. 17.1%, respectively), with a correlation between the anti-ETAR titres and the sPAP value measured by echocardiography and RHC. A recent study evaluated the prevalence of anti-ETBR antibodies in SSc patients, finding more elevated serum levels in SSc-PAH patients vs. those of healthy controls [31]; given the immunomodulatory role of ETBR_,_ it could be hypothesised that these autoantibodies can block its downstream signalling pathway, thus enhancing inflammation, but more studies are required to ascertain their exact pathogenetic role.

#### 3.2.2. Anti-Endothelial Cells Antibodies (AECA)

AECA are a heterogeneous group of autoantibodies that recognise endothelial cell proteins and molecules expressed on the endothelial cell surface, and their presence has been studied in various diseases with vascular involvement, such as CTDs, vasculitides and other non-rheumatological conditions [118,119]. Different pathophysiological effects have been observed, which include direct or indirect cytotoxicity, endothelial cell apoptosis and activation, resulting in increased leucocyte adhesiveness, the activation of coagulation and vascular thrombosis [120] and increased production of profibrotic (TGF-β) and vasoactive (ET-1) mediators [121]. Thus, it has been hypothesised that AECAs may play a pathogenetic role in PAH by inducing endothelial injury in the early stages of the disease [122]. Several studies have endeavoured to identify specific target antigens in CTDs, though with varying results. In SSc, topoisomerase I [123], centromeric protein B (CENP-B) [124], human cytomegalovirus (CMV) late protein UL94 [125] and intracellular adhesion molecule 1 (ICAM-1) [126] have been proposed as potential antigens. In SSc, AECAs have been associated with the vasculopathic manifestations of the disease, namely, PAH, digital ulcers and capillaroscopic abnormalities [127,128], suggesting a link between autoimmunity (and potential cross-reactivity to viral pathogens such as CMV) and endothelial injury. Although the association between AECAs and PAH has not been explored yet in SLE patients, a recent study has postulated that AECAs may be involved in vascular injury in the initial stages, but not in the development of overt systemic vasculitis in SLE [129]. In a cohort of MCTD patients with PAH, a significant correlation was found between the serum AECA levels and both the TM and vWF levels, suggesting that the presence of AECAs and endothelial cell activation may play a role in the development of PAH and in the maintenance of the obliterative vascular process in MCTD patients [88].

#### 3.2.3. Anti-PDGFR Antibodies

The platelet-derived growth factor (PDGF) contributes to the pathogenesis of SSc by activating the proliferation of fibroblasts and SMCs via two tyrosine-kinase receptors, PDGFRα and PDGFRβ [130]. PDGFR signalling was found to be upregulated in SSc, with a positive feedback mechanism driven by TGF-β [130,131]. There is much evidence confirming the presence of antibodies against PDGFR in SSc, other CTDs and other clinical conditions [132,133,134]. Interestingly, functional studies have shown that while agonistic anti-PDGFR antibodies (i.e., antibodies able to activate PDGFR-downstream molecular signalling pathways) are highly specific for SSc, only non-agonistic anti-PDGFR antibodies have been found in other conditions (including other CTDs) and in healthy controls [135]. Although there have been no reports of an association between anti-PDGFR and clinical features in SSc patients, a recent study found that agonistic antibodies induced the proliferation and migration of human pulmonary vascular smooth muscle cells in vitro [136]; this indicates that anti-PDGFR antibodies may be involved in establishing PAH in SSc, although this has not yet been investigated in vivo.

#### 3.2.4. Anti-BMPR Antibodies

Bone Morphogenic Proteins (BMPs) belong to the TGF-β superfamily; BMPs and their receptors (BMPR) are extensively implicated in the pathogenesis of PAH. The BMPR2-signalling pathway plays a well-recognised protective role against the development of PAH, and the genetic deficiency of BMPR2 is the most common cause of hereditary PAH [1], moreover, the expression of BMPR2 is reduced in the pulmonary vasculature of non-hereditary PAH patients [137]. Recently, Xing et al. [138] found higher levels of antibodies against BMPR1A, but not BMPR2, in SLE-PAH patients vs. those of patients without PAH and healthy controls.
ijms-24-04178-t002_Table 2Table 2Autoantibodies in CTD-PAH.Autoimmunity in CTD-PAHAutoantibodies**Disease-specific autoantibodies**SScACA [23,92]anti-pc4.2 [94]Anti-U3RNP [92]Anti-Th/To [92]aPL [99,100]SLEAnti-U1RNP [97,102]aPL [103,104,107,108]Anti-SSA, anti-SSB [102]Other CTDsAnti-U1RNP [88,108]**Functional autoantibodies**Anti-ETAR and anti-AT1R [116,117]Anti-ETBR [31]AECA [88,127,128]Anti-PDGFR [136]Anti-BMPR [138]ACA, anti-centromere antibodies; AECA, anti-endothelial cells antibodies; aPL, antiphospholipid antibodies; AT1R, angiotensin receptor type 1; BMPR, bone morphogenic protein receptor; ETAR, endothelin receptor type A; ETBR, endothelin receptor type B; pc4.2, phage clone expressing peptide 4.2; PDGFR, placental derived growth factor receptor; RNP, ribonucleoprotein.


## 4. Inflammation

Recent evidence shows that inflammation may contribute to the onset and progression of PAH, especially CTD-PAH; macrophage and lymphocyte infiltrates have been found in the pulmonary vessels of CTD-PAH patients, alongside with ANA, immunoglobulins and complement fractions [101,139]. Notably, elevated serum levels of inflammatory cytokines have been found in patients with CTD-PAH, as well as IPAH [139], and their efficacy as biomarkers in these conditions has been explored in several studies (Table 3).

### 4.1. Inflammatory Cytokines

Elevated levels of TNF-α, IL-1β, IL-6, IL-8 and IL-13 have been found in the serum of lcSSc-PAH patients [85,140]. Serum **IL-6** levels were increased in MCTD patients with PAH compared to those without PAH [141], and IL-6 blockade prevented the development of hypoxia-induced PH in murine models [142]. In a recent study, elevated IL-6 levels were identified as a strong predictor of mortality in SSc patients with pulmonary involvement (including ILD), with the highest levels being found in patients affected by PH-ILD [143]. The IL-18-binding isoform a (IL-18BPa) was found to be more elevated in the serum of SSc patients compared healthy controls, showing a positive correlation with sPAP and mPAP [144]. Di Benedetto et al. [145] found more increased serum IL-32 levels in SSc-PAH patients vs. those of SSc patients without PAH and IPAH patients. Lastly, serum levels of type I, II and III interferon (IFN) were found to be increased in patients with SSc-PAH [146]. In another study, the levels of IL-5, IL-8 and IL-12 did not differ between SSc patients with and without PH [83]. Interestingly, interleukin serum levels appeared to vary in response to vasodilators: in a small cohort of SSc-PAH patients, a significant decrease in the serum levels of IL-2, IL-6, IL-8 and IFN-γ was observed after 12 months of treatment with bosentan [147]. Increased levels of the acute phase response protein pentraxin-3 (PTX3) were found in patients with PAH, especially CTD-PAH, vs. those of healthy controls and CTD patients without PAH, without any correlation with BNP or CRP levels, identifying PTX3 as an early marker of PAH, especially in CTDs [148]. Recently, soluble markers of B cell activation were found to be elevated in the serum of SSc patients; in particular, a correlation was shown between serum levels of B-cell activating factor (BAFF), NT-proBNP levels and forced vital capacity (FVC)/DLCO ratio in lcSSc-PAH patients, although, overall, serum BAFF levels were not significantly higher in SSc patients with PAH vs. those without PAH [149].

### 4.2. Chemokines

Chemokines are a protein family whose main role is leukocyte attraction and activation; several pieces of evidence support their role in sustaining inflammation in CTD-PAH. Serum C-X-C motif ligand 4 (CXCL4) levels were elevated in SSc patients with both PAH and ILD [150], whereas high serum levels of CXCL16 were found in SSc-PAH patients [151]. In a cohort of SSc patients, C-C motif ligand 20 (CCL20) levels correlated directly with the mPAP values measured by RHC; a positive correlation was also found between CCL20 levels and the presence of primary biliary cholangitis [152], indicating a common pathogenesis between this rare manifestation and PAH in SSc. Moreover, the CCL21 serum levels were found to be elevated in the serum and lung tissue of patients with SSc-PAH [153,154].
ijms-24-04178-t003_Table 3Table 3Biomarkers of inflammation in CTD-PAH.Inflammation Molecules in CTD-PAHBiomarkers**Inflammatory cytokines**TNF-α, IL-1β, IL-8, IL-13 [85,140]IL-6 [85,140,141]IL-18BPa [144]IL-32 [145]IFN type I, II and III [146]PTX3 [148]BAFF [149]**Chemokines**CXCL4 [150]CXCL16 [151]CCL20 [152]CCL21 [153,154]BAFF, B-cell activating factor; CCL, C-C motif ligand; CXCL, C-X-C motif ligand; IL, interleukin; IFN, interferon; PTX3, pentraxin 3; TNF, tumour necrosis factor.


## 5. Cardiac Dysfunction

Vascular changes occurring in the lungs increase PVR, ultimately leading to right ventricular failure. Thus, right heart dysfunction is a late finding in the natural history of PAH. Serum biomarkers of myocardial damage and remodelling have been studied extensively in PAH as they are affordable, easy-to-use and reliable indices (Table 4); however, they are neither useful in discriminating PAH from other PH groups, nor are they specific for pulmonary hypertension, as their increase is common in many other medical conditions involving heart dysfunction, including primary heart involvement in CTD.

### 5.1. Natriuretic Peptides

Natriuretic peptides are well-recognised markers of right ventricular dysfunction in PH. They are secreted mainly by the myocardium, but also by the kidneys and the brain in response to volume overload, thus playing an important role in the homeostatic regulation of blood volume through their diuretic, natriuretic, kaliuretic and vasodilatory actions. Atrial natriuretic peptide (ANP) and brain natriuretic peptide (BNP) are the main hormones of the natriuretic peptide system; ANP is released by atrial myocytes and BNP is secreted by the ventricles.

Although ANP has been studied as a biomarker in PH, with limited results due to its complicated detection methods, there is evidence that ANP and its precursor, NT-proANP, might have a prognostic value for cardiac involvement in SSc, including PH [155,156].

Most of the available data on natriuretic peptides in PAH focus on BNP and its terminal fragment NT-proBNP. Compared to BNP, NT-proBNP has a longer half-life and it is, therefore, more stable and easier to measure. It should also be noted that it is secreted by the kidneys, thus its reliability as a biomarker depends on the renal function. BNP has a shorter half-life, but it responds rapidly to ventricular overload and does not undergo renal elimination [157]. There is plenty of evidence showing a correlation between the haemodynamic and functional parameters and the ventricular natriuretic peptides in group 1 PH [158,159,160,161], and specifically, in SSc-PAH. Both BNP and NT-proBNP can be used in the risk stratification of PAH patients according to the most recent guidelines [1]; in SSc-PAH, both peptides correlate with mPAP [162], while the NT-proBNP levels also showed a correlation with the PVR value, right atrial pressure and cardiac index in SSc-PAH [163]. Interestingly, comparing PAH subgroups, higher NT-proBNP levels were found in SSc-PAH than those in IPAH, independent of the mPAP values [164]. Overall, both peptides have a predictive value for PAH diagnosis and mortality. Based on this evidence, NT-proBNP was included in the DETECT algorithm for the identification of PAH in SSc [23].

### 5.2. Cardiac Troponin

Cardiac troponin (cTn) is the biomarker of choice for the detection of myocardial injury and for the diagnosis of myocardial infarction [165]; high-sensitivity (hs)-cTn assays are recommended for routine clinical use [165]. Besides its central role in the diagnosis and management of patients with acute coronary syndromes [165], cTn can be elevated due to a variety of cardiac and extracardiac aetiologies [166,167], and regardless of the cause, cTn increase (even below the 99th% of upper reference limit (URL) [168]) portends a poor prognosis. cTn can be elevated in patients with PAH, and specifically, with CTD-PAH; several mechanisms have been proposed for cTn elevation in patients with PAH, including myocardial ischaemia and necrosis from acute increase in the right ventricle afterload or myocardial stress from increased wall tension in a context of volume and/or pressure overload [169,170]. Moreover, cTn may also be elevated in patients with CTD due to multifactorial left ventricular [171] and kidney involvement [165].

A landmark study by Torbicki et al. [172] showed that cTnT was detectable in 14% of patients in a cohort of 56 individuals with PAH or CTEPH. The patients with detectable cTnT had higher heart rate, lower mixed venous oxygen saturation, higher serum NT-proBNP levels and a shorter 6MWD, as well as a lower survival rate compared to those without. In a study comprising 55 patients (20 with PAH, 30 with CTEPH and 5 with ILD), increased hs-cTnT levels were detected in 90% of cases, despite only 27% of the enrolled patients having a myocardial injury. In this study, hs-cTnT was associated with death and advanced WHO-FC, systolic right ventricle dysfunction and impaired 6MWD [169]. Eggers et al. [170] found that only hs-cTnI was independently associated with a worse survival in 56 patients with PAH. Regarding CTD-PAH specifically, a study comprising 33 patients with this condition showed that hs-cTnT was able to risk-stratify patients with CTD-PAH for future heart failure hospitalisation and death [173]. A 2015 study first investigated cardiac biomarkers in SSc and although only 13 patients had associated PAH documented on RHC, increased hs-cTnT levels were independently associated with a higher risk of PAH in the multivariable analysis. A more robust and recent study comprising 675 patients with SSc found that hs-cTnT was associated with an increased risk of death and PAH in the univariate analysis, although it did not add relevant information when it was integrated into models with clinically relevant variables (possibly due to the small number of patients who developed PAH, *n* = 39) [174]. Therefore, data on the prognostic role of hs-cTn in patients with CTD-PAH are still limited and controversial. Further studies in larger cohorts of CTD-PAH patients, analysing different hs-cTn assays with sex-specific 99^th^% URL and applying the new PH definition from the latest guidelines [1] are necessary to ascertain the role of this biomarker in this specific setting.
ijms-24-04178-t004_Table 4Table 4Biomarkers of right heart dysfunction in CTD-PAH.Biomarkers of right heart dysfunction
ANP [155,156]BNP/NT-proBNP [23,162,163,164]Cardiac Troponin [173,174]ANP, atrial natriuretic peptide; BNP, brain natriuretic peptide; NT, N-terminal.


## 6. Discussion and Conclusions

PAH is one of the most serious complications in patients with CTDs, in particular, SSc. CTD-PAH is the result of a complex interplay among endothelial dysfunction and vascular remodelling, autoimmunity and inflammatory changes, which ultimately leads to right heart dysfunction and failure.

Improving the screening strategies for the early detection of CTD-PAH is paramount; consistent evidence is available for SSc, showing that early diagnosed PAH patients have less severe haemodynamic impairment and better survival rates compared with those of non-screened patients [175].

In recent years, several studies have focused on the detection of biomarkers of PAH in CTD patients. Only a few molecules mentioned in the present study are already in use in clinical practice to assess the likelihood of PAH in SSc patients as part of the DETECT algorithm, namely, ACA, UA and BNP. However, these routinely used biomarkers fail to identify the early pathogenetic changes in CTD-PAH. Some recent studies have yielded promising results, with the identification of several biomarkers that may play a key role in the pathogenesis of CTD-PAH. Along with autoimmunity, which helps to predict the risk of developing PAH according to the presence of specific functional and non-functional autoantibodies, inflammation and vascular remodelling have also been explored as the two pivotal driving mechanisms in the natural history the disease. Thus, autoantibodies, inflammation molecules, vasoactive mediators and connective tissue products have been proposed as potential biomarkers of early disease-related damage in CTD-PAH; these molecules are not yet validated for use in clinical practice due to the lack of strong supporting evidence and standardised detection methods. Several studies have focused on the role of ET-1 and functional autoantibodies, particularly, anti-ETAR and anti-AT1R, as biomarkers of CTD-PAH and PAH in general, and their serum levels are occasionally measured for clinical purposes in referral centres. Thus, these biomarkers appear to be the most promising for the near future as part of routine laboratory work-up for CTD-PAH. Nevertheless, larger prospective studies are needed to ascertain the value of all of the aforementioned biomarkers as potential screening tools in everyday clinical practice toward improving the early diagnosis and risk stratification of CTD-PAH patients.

There are limited data on serum biomarkers in subtypes of PAH other than CTD-PAH. The comparison between CTD-PAH and IPAH is possible for some of the aforementioned molecules (Appendix A), but many biomarkers have been studied only in CTD cohorts (thus, comparing patients with CTD-PAH vs. CTD without PAH), and others (e.g., autoantibodies) have not been explored in other types of PAH due to the unique pathogenesis of autoimmune diseases.

Moreover, several studies have shown that PH subtypes other than group 1 may occur in the setting of CTDs, and discriminating between the different aetiologies can pose a challenge for clinicians. For instance, in SSc, groups 2, 3 and 4 PH must be considered for the differential diagnosis once PAH is suspected [176]. Differentiating SSc-PAH from PH-ILD is particularly difficult due to the high prevalence of ILD in SSc, and to the fact that both entities show a pre-capillary pattern in RHC; however, since the prognosis and the treatment of these two conditions differ radically, it is of the utmost importance to make the correct diagnosis at the baseline. Another challenge lies in the ability to distinguish between PAH and PH due to left heart disease; although the wedge pressure evaluation is helpful in the differential diagnosis, in some patients with CTD, in particular SSc, the correct classification of the disease may be more complex, given the high frequency of subclinical myocardial involvement and the lack of a validated cut-off value after the fluid challenge. Regarding PVOD, it may be useful to identify specific biomarkers of arteriolar vs. venular damage, though it bears noting that the latest guidelines do not classify PVOD as a separate entity, but it is defined in group I as “PAH with features of venous/capillary involvement”. This underlines that both the venous and arterial systems may be involved in PAH patients with PVOD features, especially CTD patients [177]. To our knowledge, no validated biomarker has the discriminatory power to segregate PAH from other types of PH in CTD patients, and future studies are necessary to meet this clinical need.

## Figures and Tables

**Figure 1 ijms-24-04178-f001:**
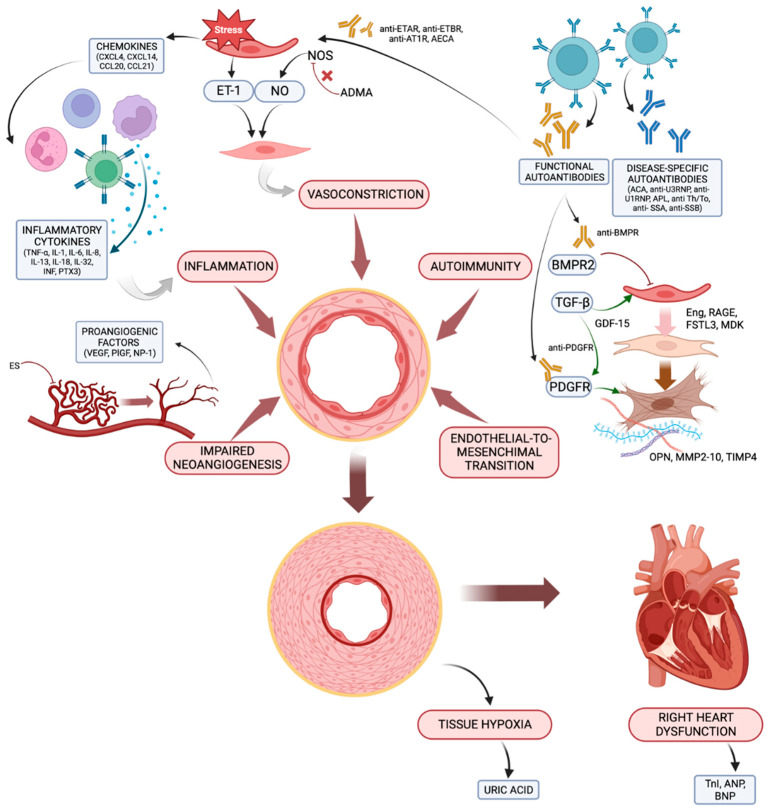
The main pathogenetic mechanisms leading to CTD-PAH and the associated biomarkers. ACA, anti-centromere antibodies; ADMA, asymmetric dimethylarginine; AECA, anti-endothelial cells antibodies; aPL, antiphospholipid antibodies; ANP, atrial natriuretic peptide; AT1R, angiotensin receptor type 1; BMPR, bone morphogenic protein receptor; BNP, brain natriuretic peptide; CCL, C-C motif ligand; CXCL, C-X-C motif ligand; Eng, endoglin; ES, endostatin; ET-1, endothelin-1; ETAR, endothelin receptor type A; ETBR, endothelin receptor type B; FSTL-3, follistatin-like 3; GDF-15, growth differentiation factor-15; IL, interleukin; IFN, interferon; MDK, midkine; MMP, matrix metalloproteinase; NO, nitric oxide; NP-1, neuropilin-1; OPN, osteopontin; PDGFR, placental derived growth factor receptor; PlGF, placental growth factor; PTX3, pentraxin 3; RAGE, receptor for advanced glycation end products; RNP, ribonucleoprotein; TIMP, tissue inhibitor of metalloproteinase; TNF, TnI, troponin I; tumour necrosis factor; VEGF, vascular endothelial growth factor. Created with BioRender.com.

## Data Availability

Not applicable.

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
