# Peer review of "Serum Biomarkers in Connective Tissue Disease-Associated Pulmonary Arterial Hypertension"

_ijms, 2023, doi:10.3390/ijms24044178_

Round 1

Reviewer 1 Report

In the manuscript submitted to IJMS entitled “Serum Biomarkers in Connective Tissue Disease-associated Pulmonary Arterial Hypertension”, the authors summarize a series of potential biomarkers of CTD- PAH and classify them by relevant pathophysiological mechanisms. The manuscript is informative and novel.

The reviewer has several suggestions for improvement.

1. Please proofread the language

2. What is the meaning of the bold in the manuscript? If they are abbreviations, please align all abbreviations in the manuscript and make them bold

3. Table 1 borders are not consistent in thickness

4. Parts 1-4 can be summarized briefly in a figure/table at the end, respectively

5. the conclusion part is too brief, suggest adding an outlook part.

6. The number of references in the last 3 years is low, accounting for only 13%, suggest updating the recently published references related to the manuscript

7. Uniform reference format, e.g., reference 161

Author Response

Point 1: Please proofread the language

The manuscript has now undergone extensive revisions by a native English speaker with particular experience in proofreading scientific articles. Thank you.

Point 2: What is the meaning of the bold in the manuscript? If they are abbreviations, please align all abbreviations in the manuscript and make them bold

We used the bold to highlight the first mention of each new biomarker; it has been specified in the text at the end of the introduction paragraph.

Point 3: Table 1 borders are not consistent in thickness

Table 1 has been divided into 4 tables as in point 4, the borders’ thickness has been evened throughout.

Point 4: Parts 1-4 can be summarized briefly in a figure/table at the end, respectively

We thank the Reviewer for their suggestion. Table 1 has been divided into 4 different tables, each one summarizing the contents of parts 1-4. Tables 1-4 are now at the end of each paragraph, respectively. Figure 1 summarizes the molecular pathways and biomarkers involved in CTD-PAH and how they interact with each other; hence the reason why we opted to keep it as a single figure to provide a more cohesive outlook on the topic.

Point 5: the conclusion part is too brief, suggest adding an outlook part.

We have now expanded the conclusion to broach current clinical challenges and critical decision points in the diagnostic process of CTD-PAH, as well as the need for future studies to identify specific biomarkers for group I CTD-PAH, toward early diagnosis and better patient outcomes. 
A brief discussion has been added (and titled “discussion and conclusions” instead of “conclusions”) on clinical challenges and critical decision points in the diagnosis of CTD-PAH, underlining the need for future studies focused on the identification of specific biomarkers for group 1 CTD-PAH.

Point 6: The number of references in the last 3 years is low, accounting for only 13%, suggest updating the recently published references related to the manuscript.

We have increased the number of recently published references, especially in the introduction section; the articles published between 2019 and 2022 now account for 20% of the total.

Point 7: Uniform reference format, e.g., reference 161.

The reference format is more consistent throughout the manuscript. Thank you.

Reviewer 2 Report

Dear editor of IJMS   

Thanks for the invitation to review the manuscript entitled (Serum Biomarkers in Connective Tissue Disease-associated Pulmonary Arterial Hypertension).

 The review is important which discussing the potential circulating serum biomarkers candidates as biomarkers of CTD-PAH diagnosis.

In this review the authors mentioned the biomarkers for connective tissue associated PAH but they did not mentioned the levels of these biomarkers in other subtybes of PAH. It must be clear. The writing style of the manuscript is attractive, however, it has only one figure and one table. A conclusive table illustrating the behavior of the mentioned biomarkers in other subtybes of PAH is necessary. (For example: increased, decreased, no change, no study, no relation).

In table 1. No references were mentioned. Please add.

Figure 1 should allocate before table 1.

Some abbreviations have no prior descriptions for example line 161.

Line 166: Asymmetric dimethylarginine (ADMA) is …..  should be in a separate paragraph.

The authors did not mention which of the previously screened biomarkers could be a potential biomarker or specific in PAH diagnosis.

The review also lacks the discussion and future studies needed to validate novel biomarkers.  

Round 2

Reviewer 2 Report

Dear Authors

thank you so much for providing answers to my comments and suggestion. 

regarding the conclusive table that you submitted in the response letter, i think it can be added at least as a supplementary data and mention your reply as a kind of limitation. it will be useful for the readers. 

i mean this part can be mentioned as follow

Data on serum biomarkers in subtypes of PAH other than IPAH are lacking. The comparison between CTD-PAH and IPAH is possible for some of the molecules (supplementary table 1), but many biomarkers have been studied only in CTD cohorts (thus, comparing patients with CTD-PAH vs. CTD without PAH); and for others (e.g., autoantibodies) the comparison is not possible, due to the unique pathogenesis of autoimmune diseases. Here is a draft of a table comparing serum levels of the mentioned biomarkers in CTD-PAH vs. IPAH.

....These molecules are not yet validated for use in clinical: please mention their name  (.....)

Author Response

Point 1: regarding the conclusive table that you submitted in the response letter, i think it can be added at least as a supplementary data and mention your reply as a kind of limitation. it will be useful for the readers. 

I mean this part can be mentioned as follow

Data on serum biomarkers in subtypes of PAH other than IPAH are lacking. The comparison between CTD-PAH and IPAH is possible for some of the molecules (supplementary table 1), but many biomarkers have been studied only in CTD cohorts (thus, comparing patients with CTD-PAH vs. CTD without PAH); and for others (e.g., autoantibodies) the comparison is not possible, due to the unique pathogenesis of autoimmune diseases. Here is a draft of a table comparing serum levels of the mentioned biomarkers in CTD-PAH vs. IPAH.

We thank the reviewer for this valuable suggestion. A supplementary material file has been created, containing table S1 as suggested. A paragraph illustrating these concepts (with reference to the corresponding table) has now been added the main text, in the discussion/conclusion section.

Point 2: These molecules are not yet validated for use in clinical: please mention their name (...) 

“These” in the text refers to “autoantibodies, inflammation molecules, vasoactive mediators and connective tissue products”; the punctuation has been slightly modified to make the connection clearer.